# Residue-Specific Epitope Mapping of the PD-1/Nivolumab Interaction Using X-ray Footprinting Mass Spectrometry

**DOI:** 10.3390/antib13030077

**Published:** 2024-09-19

**Authors:** Line G. Kristensen, Sayan Gupta, Yan Chen, Christopher J. Petzold, Corie Y. Ralston

**Affiliations:** 1Lawrence Berkeley National Laboratory, Molecular Biophysics and Integrated Bioimaging Division, Berkeley, CA 94720, USA; lkristensen@lbl.gov (L.G.K.); sayangupta@lbl.gov (S.G.); 2Lawrence Berkeley National Laboratory, Biological Systems and Engineering Division, Berkeley, CA 94720, USA; yanchen1998@lbl.gov (Y.C.); cjpetzold@lbl.gov (C.J.P.); 3Lawrence Berkeley National Laboratory, Molecular Foundry Division, Berkeley, CA 94720, USA

**Keywords:** hydroxyl radical footprinting, X-ray footprinting mass spectrometry (XFMS), epitope mapping, programmed cell death protein 1, PD-1, nivolumab, ICI

## Abstract

X-ray footprinting coupled with mass spectrometry (XFMS) presents a novel approach in structural biology, offering insights into protein conformation and dynamics in the solution state. The interaction of the cancer-immunotherapy monoclonal antibody nivolumab with its antigen target PD-1 was used to showcase the utility of XFMS against the previously published crystal structure of the complex. Changes in side-chain solvent accessibility, as determined by the oxidative footprint of free PD-1 versus PD-1 bound to nivolumab, agree with the binding interface side-chain interactions reported from the crystal structure of the complex. The N-linked glycosylation sites of PD-1 were confirmed through an LC-MS/MS-based deglycosylation analysis of asparagine deamidation. In addition, subtle changes in side-chain solvent accessibility were observed in the C′D loop region of PD-1 upon complex formation with nivolumab.

## 1. Introduction

Programmed cell death protein 1 (PD-1) and its natural ligand PD-L1 have become potent targets for cancer immunotherapy [1,2,3,4]. PD-1 is a 288 amino acid transmembrane glycoprotein that belongs to the immunoglobulin superfamily, as evidenced by its characteristic IgV domain. The PD-1 receptor is expressed on activated T cells, B cells, and myeloid cells and functions as an immune checkpoint, modulated by the interaction with PD-L1. Tumor cells expressing PD-L1 inhibit T-cell activation, which enables tumor cells to evade the antitumor immune response [1,5,6,7,8].

Nivolumab received FDA approval in 2014 as an anti-PD-1 monoclonal antibody for the treatment of melanoma [9] and, since then, has been used to treat a variety of cancers. Nivolumab binds PD-1 with an affinity of *K*d = 2.6 nM [9,10] and blocks the PD-1/PD-L1 interaction which impedes T-cell inhibition and reactivates the immune response toward tumor cells.

The NMR structure of PD-1 (P34-E150) revealed considerable flexibility in the BC loop, the C′D loop, and the FG loop [11]. The two reported crystal structures of the PD-1/nivolumab complex show residues of the BC and FG loops of the IgV domain and residues of the N-terminal loop (N-loop) involved in binding. However, the earlier crystal structure of the complex (PDB: 5GGR) [12] is missing two important N-loop residues [13] while the later crystal structure (PDB: 5WT9) [14] contains the entire N-loop of the mature protein, starting at L25. The highly flexible N-loop of PD-1 is required for binding and dominates the interaction with nivolumab despite the N-loop not being involved in the PD-1/PD-L1 interaction [14]. Molecular dynamics simulations of the PD-1/nivolumab interaction suggest a dynamic, two-step process in which the N-loop binding to nivolumab stabilizes the interface between the IgV domain and the antibody and also facilitates the rebinding of the IgV domain with nivolumab [13]. Unlike PD-1/PD-L1, the PD-1/nivolumab interaction takes place independent of glycosylation [14].

Here, we used the method of X-ray hydroxyl radical footprinting combined with mass spectrometry (XFMS) to investigate the PD-1/nivolumab interaction in solution. XFMS relies on the X-ray-mediated radiolysis of water to generate hydroxyl radicals which can covalently label solvent accessible protein side-chains [15,16] (Figure 1A). The extent of the oxidative labeling of amino acid side-chains is determined by solvent accessibility to both bulk and bound water and the side-chain’s intrinsic reactivity toward hydroxyl radicals. Under controlled irradiation conditions, the direct effect of sample irradiation is the interaction of ionizing radiation with water, while any direct impact of radiation on protein molecules is negligible at micromolar protein concentrations [16,17]. Following the labeling reaction, bottom-up LC-MS/MS is employed to identify and quantify the covalent modifications, resulting in information about the relative change in solvent accessibility at the residue level between two or more states of the protein (Figure 1B,C). Modifications are commonly observed for 16 of the 20 naturally occurring amino acids, and under aerobic conditions, hydroxyl radical labeling results in side-chain mass shifts of primarily +16, +14, +32, and +48 Da [16]. Because XFMS is complementary to other more well-known structural biology methods [15,17,18,19,20,21,22,23], we sought to both compare XFMS structural information with previous protein crystal structures, and to gain new insight into the PD-1/nivolumab interaction under solution state conditions.

## 2. Materials and Methods

### 2.1. X-ray Footprinting of Protein Samples

Lyophilized human PD-1 ectodomain (L25-Q167, GenScript, Piscataway, NJ, USA, Z03424) was reconstituted in PBS pH 7.4 (Gibco, Thermo Fisher Scientific, Waltham, MA, USA, 10010-023) to a concentration of 100 µg/mL. Full-length nivolumab at 5 mg/mL (Selleckchem, Houston, TX, USA, A2002) and reconstituted PD-1 were dialyzed against PBS to remove hydroxyl radical scavenging compounds prior to footprinting. Protein concentration after dialysis was confirmed by A_280_. Free PD-1 was prepared at 5.5 µM for footprinting while the complex mixture was prepared as a 1:1 ratio of 2.8 µM PD-1 to 2.8 µM full-length nivolumab. The complex mixture was incubated at room temperature with gentle orbital shaking for 30 min. All samples were held on ice for a few hours prior to footprinting. The samples were irradiated at the Advanced Light Source (ALS) beamline 3.3.1 using a syringe pump and capillary sample delivery method, as previously described [24]. A 5 µM Alexa 488 dose response in the presence of protein was carried out to ensure that the two samples with the same total protein concentration produced a similar Alexa dose response, as measured using the Alexa 488 rate constant [25]. The Alexa 488 rate constant for free PD-1 in PBS was 1600 s^−1^ while the Alexa 488 rate constant for the complex in PBS was 1300 s^−1^. The X-ray exposure time ranged from 250 to 1000 µs and exposed samples were collected in tubes containing methionine amide to immediately quench any secondary radical reactions. Samples were stored at −80 °C.

### 2.2. Sample Prep for LC-MS/MS Analysis

Irradiated samples and non-irradiated control samples were desalted and buffer exchanged into 50 mM ammonium bicarbonate using 0.5 mL 3K MWCO Amicon Ultra spin filters. A starting volume of 100 µL was spun down to approximately 20 µL followed by the addition of 480 µL 50 mM ammonium bicarbonate which was spun down to deadstop. The recovered concentrated sample was adjusted to a volume of 50 µL. Dithiothreitol was added to a final concentration of 5 mM and the sample was incubated at 65 °C for 30 min. The cooled sample was alkylated with iodoacetamide at a final concentration of 15 mM for 30 min at room temperature in the dark. Following reduction and alkylation, each sample was split into two 26 µL aliquots to which either trypsin/Lys–C (Promega, Madison, WI, USA) or chymotrypsin (Promega, Madison, WI, USA) was added at a 1:20 enzyme–protein ratio (*w*/*w*). Two protease digests were employed to obtain full sequence coverage. Samples were incubated overnight at 37 °C, 200 rpm for 14–16 h, after which the digestion was terminated by heating the samples to 95 °C for 10 min followed by cooling to room temperature. A total of 500 units (1 µL) of glycerol-free PNGase F (New England Biolabs, Ipswich, MA, USA) was added per sample and the mixture was incubated at 37 °C for 16 h. The reaction was terminated by adding 2% formic acid for a final concentration of 0.1%.

### 2.3. Mass Spectrometry

An Orbitrap Exploris 480 mass spectrometer (Thermo Fisher Scientific, Waltham, MA, USA) coupled to an Agilent 1290 UHPLC system (Agilent Technologies, Santa Clara, CA, USA) was used for the LC-MS/MS analysis of peptides. An InfinityLab Poroshell 120 EC-C18 column (2.1 × 100 mm, 1.9 µm particle size, 60 °C) with an initial 0.400 mL/min flow rate was used for the separation of peptides, which eluted with the following gradient: 98% solvent A (0.1% formic acid) and 2% solvent B (99.9% acetonitrile, 0.1% formic acid) initially, followed by increasing solvent B to 10% over 1.5 min, then increasing to 35% over 10 min, then increasing to 80% over 0.5 min, holding for 1.5 min at a flow rate of 0.6 mL/min, followed by a ramp back down to 2% over 0.5 min, where it was held for re-equilibrating the column to the original conditions. The mass spectrometer settings were as follows: full scan Orbitrap resolution at 60,000; AGC Target at 3.0 × 10^6^ maximum injection time after 60 ms; the top 10 intense ions were isolated for HCD fragmentation per MS scan with collision energy set to 30% and an intensity threshold at 5.0 × 10^3^; dynamic exclusion duration set at 2 s; data-dependent MS2 scan Orbitrap resolution at 15,000; AGC target at 1.0 × 10^5^; a maximum injection time after 50 ms.

### 2.4. Analysis of LC-MS/MS Data

Data analysis was performed using PMI Byos^®^ v.5.3.44 (Protein Metrics, Boston, MA, USA). Commonly observed +14, +16, +32, and +48 Da oxidation products [16] were specified as variable modifications in the database search. Carbamidomethylation was set as a fixed modification for Cys, and Asn deamidation was set as a fixed modification for peptides containing Asn residues showing >90% deamidation.

Retention-time specific MS/MS spectra showing a high degree of fragment-ion coverage were validated manually to ensure the confident assignment of residue-specific modifications. The quantification of modifications was based on the extracted ion-chromatogram peak areas of the modified and native peptides. The fraction unmodified for each peptide was calculated as the ratio of the integrated peak area of the unmodified peptide to the sum of the integrated peak areas from the modified and unmodified peptides, and the fraction unmodified was normalized against any background oxidation seen in the unexposed control sample. The fraction of unmodified protein as a function of exposure time was plotted in Origin v.2019b (OriginLab, Northampton, MA, USA) and the dose–response profiles were fitted to the first-order exponential function *y* = *e^−kt^*. The rate constant, *k*(s^−1^), is a measure of the intrinsic hydroxyl radical reactivity and the solvent accessibility of the residue, while the ratio (R) of rate constants provides a measure of the relative change in the solvent accessibility of the residue between the free and complex states of the protein [17].

## 3. Results

### 3.1. PD-1 Deglycosylation Analysis

We first analyzed the LC-MS/MS footprinting data to assess the PNGase F deglycosylation reaction and to verify the expected N-linked glycosylation sites of PD-1, since enzyme-mediated N-linked deglycosylation results in Asn deamidation. A search for the deamidation of Asn using the zero-exposure control samples showed >90% deamidation for N49, N58, N74, N102, and N116. The high degree of deamidation of N102 was unexpected, since this residue has not previously been identified as a site of N-linked glycosylation [4]. N102 is highly conserved but has the motif NGR which does not conform to the consensus sequence N-X-S/T [26] associated with N-linked glycosylation. Asn can undergo spontaneous nonenzymatic deamidation [26], and it has been shown that a high deamidation rate is possible after a typical overnight proteolytic digestion when Asn is followed by Gly in the peptide sequence [27], which is indeed the case here for residue N102. We observed a >90% deamidation of N102 in both the trypsin/Lys-C and the chymotrypsin zero-exposure samples. Asn deamidation was subsequently set as a fixed modification for oxidative modification searches involving peptides containing N102 and the four N-linked glycosylation sites.

### 3.2. Epitope Mapping Using Hydroxyl Radical Footprinting of the PD-1/Nivolumab Complex

Footprinting results showed extensive labeling across the PD-1 sequence; however, the oxidative modifications selected for inclusion in this study were limited to those for which a confident assignment could be made based on a high degree of fragment-ion coverage. The LC-MS/MS analysis of the protease-digested samples produced residue-specific hydroxyl radical reactivity rate constants for each state (Appendix A and Appendix A) and the ratio of those rate constants in turn revealed the relative change in solvent accessibility for a particular residue (Figure 2).

The PD-1/nivolumab epitope footprinting results show a high degree of protection of N-loop residues as well as FG loop residues (Figure 2 and Figure 3). L25, D26, P28, D29, P31/W32, A129/P130, P130, and K131 all show a greater than three-fold decrease in solvent accessibility, with P28 and D29 being protected in the complex to such an extent that no hydroxyl radical labeling was observed (Appendix A). I126 and I134 flank the FG loop, and both show a two-fold decrease in solvent accessibility.

There is no change in solvent accessibility for the footprinted AB-loop residues, and the BC-loop residues (F56-S62) did not generate oxidative modifications. F56 is not solvent exposed in either the apo structure (PDB: 3RRQ) or in the complex structure (PDB: 5WT9) [14]. The Ser, Thr, and Asn residues of the BC loop have low intrinsic reactivities toward hydroxyl radicals and seldom produce detectable products, which explains the lack of modification of the BC-loop residues.

L79, F82, and a majority of the C′D-loop residues were modified and showed a moderate decrease in solvent accessibility overall. The C′D loop was not resolved in the Tan et al. [14] crystal structure of the PD-1/nivolumab complex (Figure 3) but is stabilized and resolved in the crystal structure of PD-1 bound to pembrolizumab [30], indicating that the C′D loop is not part of the PD-1 epitope of nivolumab. The X-ray footprinting results support this conclusion given the relatively moderate change in solvent accessibility of the C′D-loop residues; however, the decrease in solvent accessibility of F82, in particular, points to allosteric conformational changes in PD-1 upon complex formation. Recent work characterizing antibodies against COVID-19 variants observed similar antibody-binding induced allosteric changes in the antigen [31].

A direct comparison of XFMS-identified residues with crystal structure residues involved in the binding interface shows excellent agreement between the two structural biology methods (Table 1). Of the 14 PD-1 residues determined to be involved in either hydrogen bonding or other atom-to-atom contacts with nivolumab residues, XFMS saw modification data and a greater than three-fold decrease in solvent accessibility for eight of those residues.

## 4. Discussion

To date, nearly a dozen monoclonal antibodies targeting PD-1 have been FDA-approved, a subset of which have been crystallized in complex with PD-1. In addition to the crystal structures of PD-1 in complex with nivolumab [14] and pembrolizumab [30], the PDB holds structures for PD-1 complexed with tislelizumab [32], camrelizumab (PDB: 7CU5), toripalimab (PDB: 6JBT), cemiplimab [33] (PDB: 7WVM), serplulimab [34] (PDB: 7E9B), and dostarlimab [35] (PDB: 7WSL). Interestingly, of these crystal structures, most interactions between the antibody and PD-1 occur on one or several loops of PD-1, including the FG, C′D, BC, and N-loops [1]. Here, we have performed the first XFMS structural analysis of a PD-1/antibody interaction, choosing the full-length nivolumab in complex with the ectodomain of human PD-1 to validate the method against the corresponding crystal structure. The XFMS mapping of the solution state PD-1 epitope of full-length nivolumab showed two strongly protected regions at the N-loop and the FG loop, confirming the binding interface determined from the crystal structure of the PD-1/nivolumab interaction. In addition, XFMS showed moderate protections corresponding to a region encompassing the C′D loop, which is the loop recognized by pembrolizumab. The crystal structure of PD-1/pembrolizumab showed a binding interface consisting of residues V64, N66, Y68, Q75, T76, D77, K78, P83, E84, D85, R86, S87, Q88, and P89, and the XFMS data showed a moderate protection of Q75, L79, F82, F82/P83, E84/D85/R86, and P89 in that region. While XFMS cannot distinguish between protections due to conformational changes in a protein versus protections due to interaction with a binding partner, it is nonetheless interesting to note that nivolumab and pembrolizumab have shown partial complementary binding, despite the proximity of their respective binding sites on PD-1 [14]. The subtle changes in solvent accessibility in the C′D region could be due to a slight stabilization of the loop, which would affect the complementarity of binding of the two antibodies.

Since structural methods each have their strengths and limitations, they are often used together to give a full picture of protein–protein interactions. Crystal structures give detailed atomic models, for instance, yet disordered regions are either not present in the structure or the crystal matrix will force loops into static conformations not necessarily representative of the solution-state structure. XFMS data, while sometimes challenging to interpret in the absence of other structural data, complements high-resolution structural models with a nuanced picture of changes in side-chain solvent accessibility. In addition, XFMS has the ability to report on structural features that are too flexible to be captured by crystallography, and can help to distinguish between biologically relevant interfaces and crystal contacts [36,37,38]. In summary, we have demonstrated here the utility of XFMS in mapping a loop-defined epitope on PD-1 in the solution state, highlighting the need for the integration of structural methods to fully characterize antibody–antigen interactions.

## Figures and Tables

**Figure 1 antibodies-13-00077-f001:**
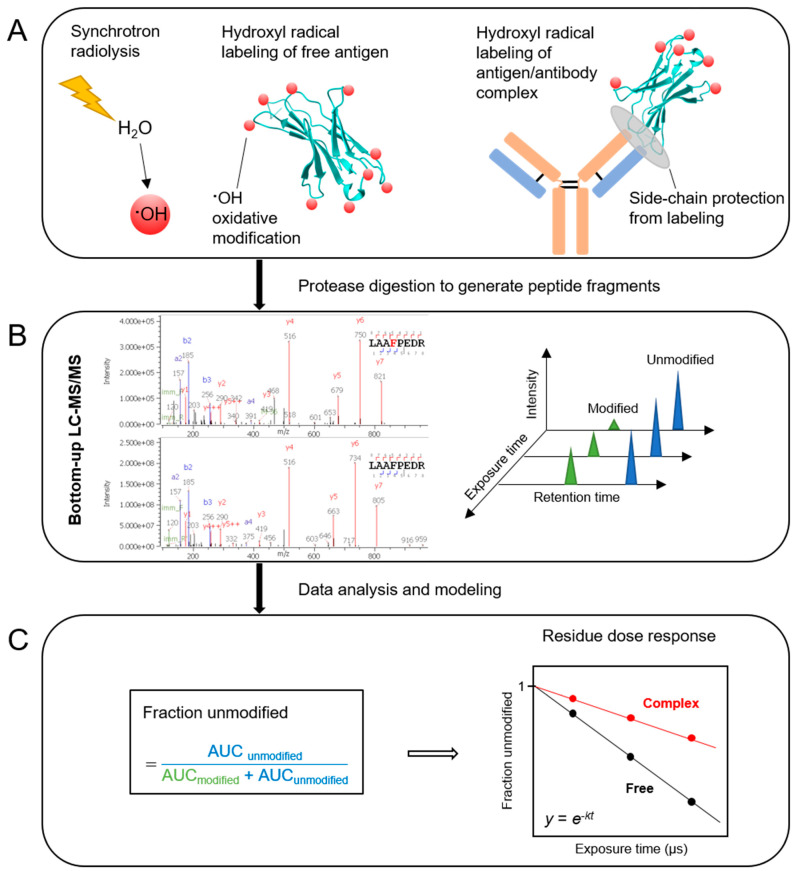
Schematic representation of the X-ray hydroxyl radical footprinting mass spectrometry workflow. (**A**) Synchrotron irradiation of a dilute protein sample in an aqueous, buffered solution produces hydroxyl radicals as a result of radiolysis of water, and hydroxyl radicals covalently label solvent-exposed side-chains if they are produced in proximity to the side-chain. The interface between antigen and antibody provides protection from labeling which leads to a different oxidative footprint from that of the free antigen. (**B**) Bottom-up LC-MS/MS analysis of protease-digested samples produces chromatograms of modified and unmodified peptides for each exposure time. (**C**) Fraction unmodified, calculated on the basis of the peak areas under the curve (AUC), is plotted as a function of exposure time. The dose response plot is fitted to a first-order exponential equation which generates the hydroxyl radical reactivity rate constant, *k*(s^−1^). The ratio of hydroxyl radical reactivity rate constants is independent of the intrinsic reactivity of the residue and the ratio therefore represents the relative change in solvent accessibility of a particular residue.

**Figure 2 antibodies-13-00077-f002:**
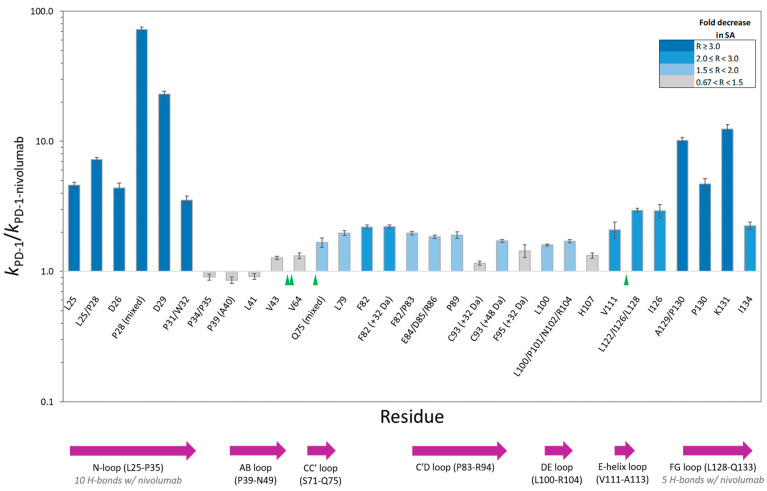
PD-1 residue-specific change in solvent accessibility (SA) upon PD-1/nivolumab complex formation. The height of each column corresponds to the ratio (R) of the hydroxyl radical reactivity rate constant of free PD-1 to the hydroxyl radical reactivity rate constant of PD-1 bound to nivolumab. Unless noted, the residue modification represents a hydroxylated product with a +16 Da mass shift. Co-eluting, modified peptide isomers are shown as mixed modifications. The bar chart color scheme reflects changes in solvent accessibility between free PD-1 and the PD-1/nivolumab complex. Gray-colored bars indicate the modification observed, but minimal change in solvent accessibility. Error bars represent the SD of the ratio [28]. The sequence locations of N-linked glycosylation sites and PD-1 structural loops are indicated with green triangles and purple arrows respectively.

**Figure 3 antibodies-13-00077-f003:**
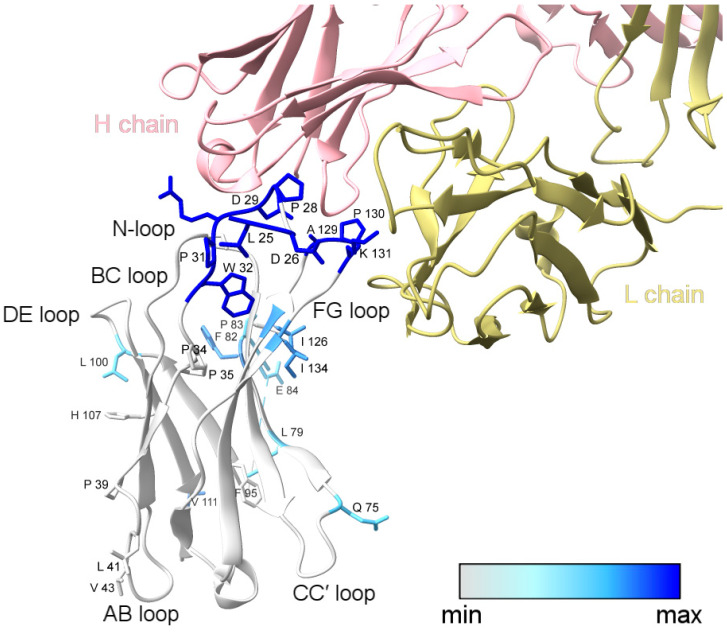
XFMS data mapped onto the structure of PD-1 bound to nivolumab-Fab (PDB: 5WT9). The blue color gradient for footprinted PD-1 residues indicates the change in solvent accessibility upon complex formation, with deep blue representing a greater than three-fold decrease in solvent accessibility. Labeled side-chains in gray showed modification but minimal change in solvent accessibility. The modified residues D85, R86, P89, and C93 could not be visualized in the structural model due to the missing C′D loop. This figure was prepared with ChimeraX [29].

**Table 1 antibodies-13-00077-t001:** PD-1 residue-level comparison of XFMS and crystallography.

PD-1 Residues Showed a Greater than Three-Fold Decrease in Solvent Accessibility when PD-1 Is Bound to Nivolumab as Determined by XFMS	PD-1 Crystal Structure Residues Determined to Contribute to Contacts in the PD-1/Nivolumab Interface [14]
N-loop L25, D26, P28, D29, P31/W32 FG loop A129/P130, P130, K131	N-loop L25(2), D26, S27(1), P28(1), D29(1), R30(4), P31 BC loop T59(1), S60 FG loop L128, A129(1), P130(1), K131(2), A132(1)

Numbers in parenthesis indicate the total number of hydrogen bonds.

## Data Availability

Raw LCMS data are available from the corresponding author, C.Y.R., upon reasonable request.

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
