# Peer review of "Residue-Specific Epitope Mapping of the PD-1/Nivolumab Interaction Using X-ray Footprinting Mass Spectrometry"

_2073-4468, 2024, doi:10.3390/antib13030077_

Round 1

Reviewer 1 Report

Comments and Suggestions for Authors

Kristensen et al. use hydroxyl radicals generated by synchrotron radiolysis of water molecules coupled with mass spectrometry (X-ray footprinting mass spectrometry or XFMS) to identify amino acid residues on the immune checkpoint pathway molecule PD-1 that are protected from covalent modification by these reactive species due to the binding of the monoclonal antibody and clinical therapeutic agent nivolumab. The method is of interest in the effort to increase understanding of the subtleties in the structural dynamics associated with the  PD-1/nivolumab interaction and as an illustrative case study of the sorts of information that this method can provide.

The manuscript is clearly written, and the experiments appear to have been performed with appropriate rigor. I found the figures and tables to be reasonably designed and informative.

I ask the authors to respond to the following comments or questions.

1. Why do the authors think that 6 of the 14 contact residues identified by crystallography did not exhibit reduced covalent modification rate constants (by three-fold or more) by hydroxyl radicals?

2. In general, what factors might be expected to determine whether an antibody will protect a contact residue on the antigen (to the extent regarded as significant) from covalent modification by the hydroxyl radicals?

3. How does the scale/rate of hydroxyl radical generation affect the extent to which antigen residues appear to be protected or not?

4. Nivolumab binds PD-1 with a nanomolar affinity. How does the intrinsic (single-site) affinity of the antibody for the antigen of interest influence the ability of that antibody to limit solvent accessibility as defined by XFMS? Is there an affinity below which this method becomes unreliable.

5. Has this particular form of footprinting been compared with any other methods of footprinting using alternative reactive chemical species that might be easier to generate for most investigators? 

6. If such comparisons have not been done, can we know if this method, which might be limited by access to a synchrotron source, is the best?

7. Can the authors identify any insights from the XFMS results that would have implications for the design of therapeutic agents targeting PD-1 or other antigens?

Author Response

Comments 1. Why do the authors think that 6 of the 14 contact residues identified by crystallography did not exhibit reduced covalent modification rate constants (by three-fold or more) by hydroxyl radicals?

Response 1: As far as S27, T59, and S60 are concerned, radiolytic oxidation of Ser and Thr residues seldom produces stable products detectable by LCMS due to their inherent chemistry when undergoing hydroxyl radical attack (Xu G & Chance MR, Chem. Rev. 2007). We did not see modification data for R30 from either the trypsin or the chymotrypsin digestions. L128 did show modification at the single-residue level from peptides in the free PD-1 sample, however, the corresponding peptides in the complex sample produced uncertain LCMS data which in turn made accurate quantification not possible. L128 appears in the mixed +16 Da modification L122/I126/L128 which is a result of the chromatographic co-elution of three peptide isomers, i.e. peaks with inadequate baseline separation. The mixed modification showed a fold decrease in solvent accessibility of 2.96 and it is difficult to precisely ascertain to which extent L128 is driving that relatively large decrease in solvent accessibility. There was at best inconclusive LCMS data related to A132 and as such did not rise to the quality level needed for quantification. In the article text, we have noted that serine and threonine residues do not generally produce oxidative modifications (page 7, lines 205-207). In Figure 2 we note the residues involved in mixed modifications, and we also note elsewhere in the text that we limited the analysis to residues for which a confident assignment could be made, excluding those with low signal to noise (page 5, lines 169-172). We are happy to add additional text to the paper if the editors feel it is warranted. 

Comments 2. In general, what factors might be expected to determine whether an antibody will protect a contact residue on the antigen (to the extent regarded as significant) from covalent modification by the hydroxyl radicals?

Response 2: The main factor is the exclusion of water. Hydroxyl radical footprinting can help distinguish between the solvent accessible surfaces defined by the residue side chains.

It is the solvent accessibility of the side chain after complex formation that will determine to which extent a particular residue will be fully, partially, or not protected from covalent modification by hydroxyl radicals. The hydroxyl radical is similar in size to water which makes it an excellent probe of the solvent accessible surface area of the side chain. In some cases, X-ray hydroxyl radical footprinting will also pinpoint bound waters in the interior of a protein, as X-rays will interact with the bound water and produce hydroxyl radicals at the site of the bound water. 

Comments 3. How does the scale/rate of hydroxyl radical generation affect the extent to which antigen residues appear to be protected or not?

Response 3: The rate of production of hydroxyl radicals (through radiolysis of water) is proportional to the flux density of the X-ray beam, but the extent of hydroxyl radical generation does not affect the protection factor. In other words, a low rate of generation of radicals (due to low flux density beams) requires longer exposure times to generate quantifiable side chain modifications, and a high rate of generation of hydroxyl radicals will require shorter exposure times. But because the protection factor is a ratio of the hydroxyl radical reactivity rate constants between complex and free samples, the resulting map of protections will be the same. However, we should note that longer exposure times can perturb the protein’s structural integrity by secondary (unwanted) radical reactions. For best practice, we use high flux density and short exposure time to quantify side chain modification. 

Overall, in order to assess changes in residue solvent accessibility, it is necessary to achieve a steady-state concentration of hydroxyl radicals high enough to mediate modification of different solvent exposed amino acid residues that can differ by three orders of magnitude in terms of their intrinsic reactivity towards hydroxyl radicals. The goal is to produce modification data on as many solvent exposed residues as possible while keeping the overall degree of modification low which is achieved through carefully controlled exposure conditions. It is critical to not over-oxidize a sample as this could lead to not only perturbations of the native structure but it could also affect the accurate quantification of modifications since the loss of the native species is assumed to adhere to pseudo-first-order reaction kinetics.

Comments 4. Nivolumab binds PD-1 with a nanomolar affinity. How does the intrinsic (single-site) affinity of the antibody for the antigen of interest influence the ability of that antibody to limit solvent accessibility as defined by XFMS? Is there an affinity below which this method becomes unreliable.

Response 4: The results from a footprinting experiment will reflect the average species in solution at the instant the sample undergoes microsecond X-ray exposure. Accordingly, the stronger the affinity of the antibody for the antigen, the larger the portion of molecules in the bound state and in turn the stronger the signal from the bound state. Generally, affinities in the micromolar range are amenable to the method, though this will depend on the data quality.

Comments 5. Has this particular form of footprinting been compared with any other methods of footprinting using alternative reactive chemical species that might be easier to generate for most investigators? 

Response 5: Several reviews cover the different aspects of each method. A comprehensive review is given in Xu & Chance (Chem Rev. 2007). A more recent and abbreviated review can be found in Ralston & Sharp (Antibodies, 2022). In general, using X-rays to generate hydroxyl radicals obviates the need to add external reagents, such as hydrogen peroxide, which is necessary in the FPOP method and the Fenton method. Each form of footprinting has its advantages and disadvantages.

Comments 6. If such comparisons have not been done, can we know if this method, which might be limited by access to a synchrotron source, is the best?

Response 6: X-ray footprinting is one of several alternatives available for structural analysis of proteins in solution. Specifics of the project will determine which method might be best depending on user accessibility logistics and protein sample requirements.  For instance, one major advantage of  X-ray footprinting of proteins is that it relies solely on the radiolysis of water to generate hydroxyl radicals, hence there are no hydroxyl-generating additives present in solution and combined with a low oxidation regime, the native conformation of the proteins can confidently be maintained during the footprinting experiment. This is especially important for proteins sensitive to hydrogen peroxide, a reagent used in several of the other footprinting methods.  One major bottleneck is accessing the national lab’s user facility at the synchrotron source. Recently, several modes of user accessibility (including mail-in user) have significantly boosted throughput of the overall process. In this paper, we endeavored to directly compare X-ray footprinting with crystallography in order to specifically validate the X-ray footprinting method.  

Comments 7: Can the authors identify any insights from the XFMS results that would have implications for the design of therapeutic agents targeting PD-1 or other antigens?

Response 7: Local or long-range conformational changes in the antigen as a result of binding to the antibody that may not be apparent when inspecting a static crystal structure of the complex, especially in the case of flexible loops that show poor or missing electron density. Paratope mapping using XFMS analysis would be expected to be able to reveal similar subtle conformational changes distant from the binding interface. In this particular study, we do think it is interesting to note that the subtle solvent accessibility changes in the C′D region might be due to a slight stabilization of the loop when nivolumab binds, which would affect the complementarity of two antibodies binding to this region at once. 

Reviewer 2 Report

Comments and Suggestions for Authors

This is a well-written scientific paper with good quality of data. Although the results are not new regarding the epitope of PD-1 binding site, the approach X-ray footprinting with high resolution mass spectrometer did complementarily support data obtained from crystal structure. Only a few general questions as the following.

1. When using iodoacetamide to alkylate free sulfhydryl groups, the concentration of iodoacetamide was pretty high (15 mM) while protein concentration was around 2.8 uM. Was there overalkylation observed on methionine? Was the overalkylation interfere with the hydroxyl radicals? 

2. What was the advantage to identify glycosylation sites on PD-1? Why not examine the glycopeptide forms directly with Byos software? What was the reason to examine the glycosylation sites of PD-1 in the first place?

3. From the crystal structure data, L128 in FG loop was also involved in binding, while XFMS did not observe this modification on L128. Please discuss the possible reason. 

4. Was wondering why a review paper from McKenzie-Coe et al. in 2022 Chemical Reviews was not cited. 

Author Response

Comment 1. When using iodoacetamide to alkylate free sulfhydryl groups, the concentration of iodoacetamide was pretty high (15 mM) while protein concentration was around 2.8 uM. Was there overalkylation observed on methionine? Was the overalkylation interfere with the hydroxyl radicals?

Response 1: To clarify, reduction and alkylation takes place after footprinting during sample prep for LCMS and there is thus no impact on the generation of hydroxyl radicals during the footprinting experiment. The standard iodoacetamide concentration of 15 mM is in line with the reported optimal concentration for a similar peptide concentration (Suttapitugsakul S, Xiao H, Smeekens J, Wu R, Mol Biosyst. 2017). However, the use of iodine-containing alkylation agents do present a challenge in terms of unwanted side reactions and we observe carbamidomethylation of methionine. We have not explored the use of alternative alkylation agents such as acrylamide.

Comment 2. What was the advantage to identify glycosylation sites on PD-1? Why not examine the glycopeptide forms directly with Byos software? What was the reason to examine the glycosylation sites of PD-1 in the first place?

Response 2: It is definitely possible to examine released glycans using the Byos software but we were not attempting to characterize the glycans of PD-1. The reason for the deglycosylation analysis was to confirm that the N-linked glycosylation sites of PD-1 had been properly deglycosylated by PNGase F. The deglycosylation reaction is carried out post footprinting and in order to set deamidation of the involved asparagine residues as a fixed modification for the database search, it is imperative that the enzymatic reaction goes to completion or near completion. The deglycosylation analysis is merely done to confirm that the PNGase F reaction worked as expected with the side benefit that it also indirectly confirms the reported glycosylation sites of human PD-1 expressed in HEK cells. If one neglects to consider the impact of deglycosylation on the accurate mass of the associated native peptides, the quantification of residue modifications on those same peptides could suffer as a result of ignoring the deamidation mass shift.

Comment 3. From the crystal structure data, L128 in FG loop was also involved in binding, while XFMS did not observe this modification on L128. Please discuss the possible reason.

Response 3: As noted in a response to Reviewer 1, L128 showed modification at the single-residue level from peptides in the free PD-1 sample, however, the corresponding peptides in the complex sample produced uncertain LCMS data which in turn made accurate quantification not possible. L128 appears in the mixed +16 Da modification L122/I126/L128 which is a result of the chromatographic co-elution of three peptide isomers, i.e. peaks with inadequate baseline separation. The mixed modification showed a fold decrease in solvent accessibility of 2.96 and it is difficult to precisely ascertain to which extent L128 is driving that relatively large decrease in solvent accessibility.

Comment 4. Was wondering why a review paper from McKenzie-Coe et al. in 2022 Chemical Reviews was not cited.

Response 4: The paper in question is a great review paper and we have now included it in the citations.

Round 2

Reviewer 1 Report

Comments and Suggestions for Authors

The authors have clearly and comprehensively responded to the questions I posed in my review of the original submission. I found their reasoning convincing and thank the authors for such detailed explanations of their experimental approach.

Reviewer 2 Report

Comments and Suggestions for Authors

Thanks a lot for the responses. All acceptable.